# Structural Insights into the Binding of Red Fluorescent Protein mCherry-Specific Nanobodies

**DOI:** 10.3390/ijms24086952

**Published:** 2023-04-09

**Authors:** Hui Liang, Zhiqiang Ma, Ziying Wang, Peiyu Zhong, Ran Li, He Jiang, Xin Zong, Chao Zhong, Xihuan Liu, Peng Liu, Jiayuan Liu, Haoran Zhu, Rui Liu, Yu Ding

**Affiliations:** State Key Laboratory of Genetic Engineering, School of Life Sciences, Fudan University, Shanghai 200438, China; 20210700102@fudan.edu.cn (H.L.);

**Keywords:** nanobody, mCherry, red fluorescent protein, crystal structure, LaM1, LaM3, LaM8

## Abstract

Red fluorescent proteins (RFPs) have broad applications in life science research, and the manipulation of RFPs using nanobodies can expand their potential uses. However, the structural information available for nanobodies that bind with RFPs is still insufficient. In this study, we cloned, expressed, purified, and crystallized complexes formed by mCherry with LaM1, LaM3, and LaM8. Then, we analyzed the biochemical properties of the complexes using mass spectrometry (MS), fluorescence-detected size exclusion chromatography (FSEC), isothermal titration calorimetry (ITC), and bio-layer interferometry (BLI) technology. We determined the crystal structure of mCherry-LaM1, mCherry-LaM3, and mCherry-LaM8, with resolutions of 2.05 Å, 3.29 Å, and 1.31 Å, respectively. In this study, we systematically compared various parameters of several LaM series nanobodies, including LaM1, LaM3, and LaM8, with previously reported data on LaM2, LaM4, and LaM6, specifically examining their structural information. After designing multivalent tandem LaM1-LaM8 and LaM8-LaM4 nanobodies based on structural information, we characterized their properties, revealing their higher affinity and specificity to mCherry. Our research provides novel structural insights that could aid in understanding nanobodies targeting a specific target protein. This could provide a starting point for developing enhanced mCherry manipulation tools.

## 1. Introduction

Since the initial extraction and identification of green fluorescence protein (GFP) from jellyfish several decades ago [1], fluorescence proteins (FPs) have proven invaluable tools for biological research owing to their exceptional properties. Besides GFP, various FPs were widely used across diverse scientific fields, such as bioimaging [2], cell tracking, protein localization studies [3], gene expression analysis [4,5], and the detection of protein-protein interactions [6]. Moreover, in the natural environment, FPs expressed by living organisms can be utilized to gauge the dynamic quantity and activity of fused proteins based on the intensity of fluorescence [7].

Compared to GFP, red fluorescent proteins (RFPs) exhibit several benefits. The most important are their longer excitation and emission wavelengths, which can penetrate deeper into tissues [8], making them a better choice for imaging in thick samples, such as living tissues or whole organisms [9,10,11,12]. Additionally, RFPs have less spectral overlap with other fluorophores commonly used in biological imaging than GFP, making distinguishing them from other signals in multiplexed imaging experiments easier. Thus, RFPs can be more easily combined with GFP or other FPs to create multicolor imaging experiments [13]. However, one challenge with RFPs is their tendency to form tetramers, which can result in false-positive binding results [14]. To address this issue, researchers have developed monomeric RFPs, which have become increasingly popular in life science research. Of the various monomeric RFPs, mCherry, developed by Shaner et al., is the most widely used [15]. This is because mCherry exhibits several desirable properties, including high brightness, excellent photostability [13], and an emission spectrum well-suited for imaging applications. Moreover, its monomeric nature ensures it does not form tetramers, reducing the likelihood of false-positive binding results.

Despite their popularity in life science research, FPs can be challenging to be manipulated. To address this problem, researchers have begun developing binders specific to FPs, which can manipulate the protein in new and exciting ways. Researchers have developed various GFP-binding molecules, including aptamers [16], αRep [17], DARPins [18], and nanobodies [19]. These binders can isolate, purify, and control GFP with greater precision and specificity, providing various tools and assays for GFP-related research [20,21,22]. Moreover, these binders can be utilized to develop new tools and assays explicitly designed for FPs. This opens up new possibilities for studying complex biological systems and processes and developing novel therapies and diagnostic tools.

Nanobodies are a novel type of protein binder and have several advantages over conventional antibodies. Nanobodies, unlike conventional antibodies, consist of a single functional heavy chain variable domain that specifically binds to antigens, making them ideal for various applications due to their small size and exceptional stability. These unique molecules were first identified in camelids [23]. One of the main advantages of nanobodies is their low molecular weight (~13 kD), which makes them easier to produce and purify at scale. Additionally, their small size allows them to penetrate tissues more effectively than conventional antibodies, enabling them to target and bind to antigens that are inaccessible to larger molecules [24]. They also exhibit high affinity and durability [25], making them suitable for various applications. Nanobodies exhibit exceptional solubility and stability, making them highly suitable for storage and utilization in various applications. In addition, they exhibit protease resistance [26] and have the potential to be modified [25], further enhancing their versatility. Furthermore, nanobodies can be engineered for specific applications, including developing biosensors and other advanced technologies [27,28]. Their simple structure makes them easy to manipulate and modify for various applications. Nanobodies have been extensively utilized in biological research, such as targeted therapy for influenza [29,30], cancer diagnosis [31], immunotherapy [32], affinity purification [33], structural determination for proteins [34], and controlling antigens in living cells [25,35]. Furthermore, nanobodies have been successfully utilized to engineer pathogen-resistant crops by rapidly and specifically modifying susceptible crop species [20].

For FP-specific nanobodies, in 2014, Fridy et al. designed a novel pipeline to generate nanobodies targeting fluorescent proteins, specifically GFP and mCherry, resulting in nanobodies with high affinity and stability [36]. The identified nanobodies for GFP have demonstrated diverse effects on its fluorescence, such as GFP enhancers and minimizers, which increase or decrease GFP fluorescence intensity [22]. They also identified six llama nanobodies against mCherry (LaMs), namely LaM1, LaM2, LaM3, LaM4, LaM6, and LaM8. Although the structures of some mCherry-specific nanobodies have been elucidated in recent years [37,38], the binding mechanism of the remaining nanobodies (LaM1, LaM3, and LaM8) to mCherry has yet to be fully resolved.

This study aimed to elucidate the binding mechanism of LaM1, LaM3, and LaM8 to mCherry by determining their respective complex crystal structures. We then identified the properties of the resulting complexes and evaluated their binding affinity to mCherry. Subsequently, we designed novel multivalent nanobodies with higher affinity based on the complex structures and assessed their improved binding affinity to mCherry.

## 2. Results

### 2.1. Protein Expression, Purification, and Characterization

To systematically study the properties of various nanobodies that specifically bind to mCherry, we employed a prokaryotic expression system to express the relevant proteins and designed multivalent nanobodies. We first expressed nanobodies LaM1, LaM3, and LaM8, by which the complex structures of these nanobodies binding with mCherry still need to be resolved. We cloned LaM1, LaM3, and LaM8 into the pSUMO vector and purified the induced SUMO-tagged proteins by His-tag affinity chromatography with Ni-NTA resin (Appendix A). After binding to the Ni-NTA resin, the target proteins with the SUMO tag were cleaved by ULP1 on the column, and the tag-removed LaM1, LaM3, and LaM8 were further purified by ion exchange or size exclusion chromatography. We also expressed and purified His6-SUMO-LaM1-LaM8 (41.9 kD), His6-SUMO-LaM8-LaM4 (41.3 kD), His10-TEV site-mCherry (29.2 kD), and His6-SUMO-DsRed (39.6 kD). The His and SUMO tags of these proteins were also removed by TEV and ULP1 protease on the column, respectively.

The whole expression and purification process was verified by SDS-PAGE (Appendix A). The molecular weight (MW) of the proteins is close to the theoretical MW (Appendix A). To obtain the nanobody-mCherry complex, we incubated nanobodies (excess) and mCherry and separated the complex by size exclusion chromatography, in which the complex was eluted in the first peak. The excess nanobodies were eluted as the second peak (Appendix A). The samples of the first peak were further used for crystal screening (Appendix A).

We also conducted MALDI-TOF/MS analysis to determine the accurate molecular weight of the recombinant proteins for verifying their suitability for downstream assays. The recombinant proteins matched the MALDI-TOF/MS result (Appendix A). All of the molecular weights derived from MS analysis (LaM1, 13.968 kD; LaM3, 14.179 kD; LaM8, 13.698 kD; mCherry, 26.693 kD) were in close agreement with the theoretical values (LaM1, 14.080 kD; LaM3, 14.291 kD; LaM8, 13.808 kD; mCherry, 26.779 kD) calculated by ProtParam on Expasy [39]. Moreover, the peak with the highest intensity observed in MS for mCherry corresponded to the larger C-terminal fragment cleaved during chromophore maturation [40] and had an m/z of 18,396. Furthermore, a peak with half the *m*/*z* value of the full-length mCherry was detected, suggesting a doubly protonated mCherry protein.

We further verified the formation of the mChery-LaM nanobodies complex by size exclusion chromatography (SEC). The shift in elution volume observed between the complex and its corresponding monomers suggests a potential binding interaction between the two components (Appendix A). Additionally, the ultra-violet absorption (UV) curve for mCherry and LaM3 (Appendix A) displayed two peaks preceding the peak for mCherry, which could suggest the formation of oligomers composed of mCherry and LaM3.

### 2.2. Crystal Structures of a Complex of mCherry and Nanobodies

Previously, the complex structures of mCherry-LaM2, mCherry-LaM4, and mCherry-LaM6 were resolved [37,38]. We thus determined the crystal structures of the unresolved complexes: mCherry-LaM1, mCherry-LaM3, and mCherry-LaM8 (Figure 1). Data describing the structures of the mCherry-LaM1, mCherry-LaM3, and mCherry-LaM8 are summarized in Table 1. The structures were deposited to PDB with ID: 8IM1 (mCherry-LaM1), 8ILX (mCherry-LaM3), and 8IM0 (mCherry-LaM8). It is plausible that the presence of both mCherry-LaM3 and (mCherry-LaM3)_2_ in equilibrium may have contributed to their non-uniform state during crystallization. This, in turn, could have significantly reduced the crystal resolution of mCherry-LaM3 and resulted in an unsatisfactory R-merge value.

The crystal structures reveal that LaM1, LaM3, and LaM8 bind to different regions on the stave side of the mCherry β-barrel, with LaM1 and LaM3 binding in relative proximity with some steric hindrance. In contrast, LaM8 binds in a completely different area. Based on the interface structure between LaM1 and mCherry, all three complementarity-determining regions (CDRs) of LaM1 contribute to the affinity (Figure 2A). In contrast, LaM3 only utilizes CDR2 and CDR3 for binding (Figure 2E), and LaM8 relies on CDR1, CDR3, and a single Gln3 for binding (Figure 2I).

We analyzed the structural information pertaining to the interaction between mCherry and the nanobodies in detail. The interaction between mCherry and LaM1 involves all the CDRs of LaM1, as illustrated in Figure 2A. Figure 2B shows the hydrogen and salt bonds between Glu176 and Lys162 of mCherry and Ser55, Gly57, Gly59, and Arg60 of CDR2 in LaM1. Moreover, Glu160 and Lys178 of mCherry form hydrogen bonds with Glu33 and Asn34 in CDR1 of LaM1, while a salt bond is present between Lys178 and Glu33. Figure 2C demonstrates a short hydrogen bond between Lys178 of mCherry and Ser104 in CDR3 of LaM1. Glu153 of mCherry contributes to the affinity by binding to Thr31, Glu33, and Tyr35 in CDR1 of LaM1, as shown in Figure 2B. Furthermore, the indole of Trp56 in CDR2 of LaM1 generates an electrostatic field that stabilizes adjacent ions, including Glu160, Glu176, and Lys178 of mCherry, leading to three ion–π interactions at the center of the interface, as depicted in Figure 2D.

CDR2 and CDR3 of LaM3 contribute to the main binding affinity to mCherry, while no residue in CDR1 binds to mCherry (Figure 2E). Ser21 and Leu124 of mCherry bind to Tyr106 in CDR3 of LaM3, and Arg125 of mCherry binds to the oxygen atom on the backbone of Tyr106 in CDR3 of LaM3 through a hydrogen bond (Figure 2F). There are multiple cation–π and π–π interactions between Arg125 of mCherry and an aromatic cage composed of Tyr106, Tyr107, and Tyr108 of LaM3, in which the π–π interaction stabilizes the CDR3, while the cation–π interaction contributes to the affinity between mCherry and LaM3 (Figure 2F). Ser54, Trp55, and Ser56 in CDR2 of LaM3 bind to Glu94 of mCherry through hydrogen bonds, and Trp55 can reinforce this binding through anion–π interaction with Glu94 (Figure 2H). Furthermore, there are hydrophobic interactions involved in CDR2, in which Leu58 and Ile59 in CDR2 of LaM3 co-build a hydrophobic environment with Val96 of mCherry (Figure 2G).

LaM8′s binding affinity to mCherry is mainly contributed by CDR1, CDR3, and one residue in the constant region. At the same time, CDR2 has a minimal direct contribution to the interface (Figure 2I). Two hydrogen bonds are formed between Ala145, Glu144 of mCherry, and Asp111, Tyr112 in CDR3 of LaM8 (Figure 2L), respectively. In addition, a hydrophobic interaction between Ala145 of mCherry and Val102 of LaM3 is observed (Figure 2L). Arg164 and Glu144 of mCherry bind to Tyr34 in CDR1 of LaM8 through hydrogen bonds, while Gly170 of mCherry binds to Arg29 in CDR1 of LaM8 in a similar way (Figure 2K). Furthermore, salt bridges are formed between Glu144 of mCherry and Arg29 of LaM8, Arg164 of mCherry, and Glu33 of LaM8, respectively (Figure 2K). Gln3 of LaM8 also binds to mCherry at Lys168 through a hydrogen bond (Figure 2J).

### 2.3. Epitope Mapping of LaM Nanobodies Binding to mCherry by FSEC

Fluorescence-detected size exclusion chromatography (FSEC) is a technique that combines size exclusion chromatography (SEC) with fluorescence detection, allowing for the analysis of the oligomeric state and stability of fluorescently-labeled proteins or protein complexes [41,42]. Due to its highly sensitive, non-destructive, and high-throughput detection capabilities, we employed FSEC to characterize the competition binding of the LaM series of nanobodies to mCherry. We first investigated the binding of a single nanobody to mCherry (Figure 3A–D).

The binding of one nanobody, which included LaM1, LaM3, and LaM8 to mCherry, shifted the peak of mCherry at 16.95 mL to peaks near 16.28 mL using the Superdex 200 increase 10/300 size exclusive column, showing that the formed mCherry-LaM1, mCherry-LaM3, and mCherry-LaM8 complexes had similar size. However, there was always a novel peak of 14.30 mL when LaM3 was added, corresponding to the (mCherry-LaM3)_2_ heterotetramer around 84 kD (Figure 3A,B,E,F). The possible mechanism is further discussed in the Discussion Section.

When we added two different nanobodies together with mCherry, the combination of LaM1 and LaM3 did not show a new peak of mCherry binding with both of them. There might be one shared epitope on mCherry that LaM1 and LaM3 can bind to, and mCherry only chooses one LaM to bind to (Figure 3A,B).

According to the results shown in Figure 3C,D, the elution volume peak shifts from approximately 16.28 mL to 15.68 mL in the presence of both LaM1 and LaM8, indicating that these two nanobodies recognize different epitopes on mCherry and do not sterically hinder each other’s binding. The fluorescence curves of mCherry with LaM3 and LaM8 (Figure 3E,F) reveal that one of the elution volume peaks for the complexes shifts from 16.37 mL to 15.79 mL, while the other shifts from 14.30 mL to 13.97 mL. These observations suggest that LaM3 and LaM8 bind to different epitopes on mCherry and that LaM8 does not interfere with the formation of a tetramer of (mCherry-LaM3)_2_ and may even contribute to the formation of a hexamer of (mCherry-LaM3-LaM8)_2_. Finally, comparing the peaks of mCherry-LaM3 with those of LaM1, LaM8, and LaM1/LaM8 (Figure 3G,H), we conclude that the addition of LaM1 does not appear to affect the formation of mCherry-LaM3-LaM8 and (mCherry-LaM3-LaM8)_2_.

Based on the FSEC results, we can infer that LaM3 and LaM1 bind to a similar epitope of mCherry. However, at the same concentration of nanobodies, LaM3 exhibits a stronger affinity for mCherry, resulting in mCherry preferentially selecting LaM3 for binding. Nevertheless, LaM3 tends to dimerize under physiological conditions, which could lead to undesirable side effects. Therefore, in the design of multivalent mCherry nanobodies, LaM1 should be given priority over LaM3.

### 2.4. Binding Kinetics and Thermal Dynamics between mCherry and Different LaM Nanobodies

To characterize the kinetics and thermodynamic parameters of the LaM nanobodies’ binding to RFPs, we first performed a Bio-Layer Interferometry (BLI) assay (Figure 4A–D and Table 2). BLI is a label-free technology that provides rapid and precise measurements of biomolecular interactions. BLI utilizes a biosensor with a biolayer that captures and immobilizes target molecules (here, His-SUMO-tagged LaM nanobodies are captured by the NTA biosensor). Changes in biolayer thickness are detected optically, allowing for the real-time monitoring of binding events. We obtained the binding affinity and kinetics parameters of LaMs to mCherry, including their association and dissociation rates, equilibrium constants, and binding affinities. The binding parameters of LaM1, LaM3, and LaM8 to mCherry or DsRed are shown in Table 2.

The K_D_s of LaM1, LaM3, and LaM8 binding to mCherry were 12.9, 30.6, and 36.8 nM, respectively, by BLI. LaM1 showed slightly better affinity (K_D_) than the other two nanobodies. The k_on_s of three monomeric LaM to mCherry were more than 9 × 10^4^/(M × s), while LaM8 showed the fastest binding with the highest k_on_ (Figure 4C). The k_dis_s of LaM1, LaM3, and LaM8 binding to mCherry were 1.17, 3.24, and 9.26 × 10^−3^/s, respectively, while LaM1 had the best k_dis_ (lowest numeric) in the three LaMs (Figure 4A). LaM3 also binded to the tetrameric DsRed protein (Figure 4D). The BLI result showed that LaM3 had a weaker binding affinity to DsRed when compared to the monomeric mCherry, with a K_D_ of 7.13 × 10^3^ nM.

Isothermal Titration Calorimetry (ITC) is widely considered the gold standard for determining the thermodynamics of molecular interactions. It measures heat changes accompanying binding reactions and provides direct, label-free, and accurate measurements of binding parameters, including stoichiometry, affinity, enthalpy, and entropy. We also used ITC to obtain precise binding parameters, and the resulting data are presented in Figure 4E–I and Table 3. The K_D_s of LaM1, LaM3, and LaM8 to mCherry were determined to be 289, 13.7, and 199 nM, respectively. In comparison, the affinity measured by BLI may not be as accurate as ITC, particularly if the dissociation curve in BLI is slow, which could result in a greater degree of error.

The ITC results demonstrated that the binding of LaM1 or LaM3 to mCherry did not interfere with the binding of LaM8 (Figure 4H,I). Specifically, LaM8 could bind to the complex of mCherry-LaM1 and the complex of mCherry-LaM3, as shown in Figure 4H,I. Interestingly, the binding of LaM1 even increased the binding affinity of LaM8 to mCherry (from K_D_ 199 nM to 54 nM) (Figure 4G,H), while the binding of LaM3 only showed a slight effect (from K_D_ 199 nM to 236 nM) (Figure 4G,I). Furthermore, the Gibbs free energy change (ΔG) was lower than −35 kJ/mol in all groups, indicating a strong tendency toward binding. As discussed in the crystal structure section, this exothermic reaction may be due to the formation of new hydrogen bonds, electrostatic interactions, Van der Waals interactions, hydrophobic interactions, or π–π interactions between LaM nanobodies and RFP.

### 2.5. Comparison of the Emission Fluorescence Spectrum of mCherry Binding with Different LaMs

Our results (Figure 4J) showed the maximum fluorescence wavelength (Em) of mCherry (616.6 nm) or mChery-LaM1 (617.2 nm), mChery-LaM3 (616.4 nm), and mChery-LaM8 (618.6 nm) complexes were kept in a minimal range from 616.4 to 618.6 nm, and the fluorescence intensity also did not significantly change. In contrast to GFP nanobodies, which may substantially alter the fluorescence intensity and spectrum of GFP [22], we found that the binding of LaMs has minimal effects on the excitation and emission spectra of mCherry. Based on the structures we have resolved, it appears that upon binding of LaMs to mCherry, the mCherry’s chromophore remains relatively unchanged. This makes LaMs valuable for researchers who want to monitor or manipulate RFPs without altering their properties. Additionally, fluorescence intensity data obtained using LaMs can be easily processed without complicated intensity adjustments. This highlights the potential of LaMs as robust and user-friendly tools for monitoring and manipulating RFPs.

### 2.6. Design and Verification of Multivalent Nanobodies Binding with mCherry

Based on the structural and binding assay data of LaM nanobodies, including those developed previously, such as LaM2, LaM4, and LaM6 [37,38], we designed several multivalent nanobodies to improve their binding affinity and selectivity. These multivalent nanobodies were then tested by BLI (Figure 5A−C and Table 4) and ITC (Figure 5D,E and Table 5) to determine their binding kinetics and thermodynamics.

Based on the crystal structure of the mCherry-LaM nanobody complex, we measured the distance between the amino and carboxyl ends of the candidate LaM nanobodies and selected appropriate combinations of nanobodies. Although the LaM family comprises six candidate nanobodies, certain nanobodies exhibit overlapping binding epitopes (e.g., LaM1, LaM2, LaM3, and LaM6), rendering them unsuitable for use in combination. Furthermore, the connection of certain nanobodies within the family may require a very long linker, making their incorporation challenging. In addition, LaM3 may lead to unnecessary multimerization. Ultimately, we verified the effectiveness of two combinations, LaM1-LaM8 and LaM8-LaM4. The distance between the C-terminus of LaM1 and the N-terminus of LaM8 is 51.1 Å, while the distance between the C-terminus of LaM8 and the N-terminus of LaM4 is 56.8 Å. Since the peptide chain length formed by each five repeated small amino acids (GGGGS) is about 17.25 Å [21,36], and 4 GGGGS is above 60 Å, we used (GGGGS)_4_ to link LaM1-LaM8 and LaM8-LaM4.

The association rate constants (k_on_) of LaM1-LaM8 and LaM8-LaM4 were found to be 2.96 × 10^5^/(M × s) and 1.73 × 10^5^/(M × s), respectively, while the dissociation rate constants (k_dis1_) were measured to be 5.52 × 10^−4^/s and 2.39 × 10^−4^/s, respectively. The calculated K_D_ values (1.87 and 1.38 nM) using biolayer interferometry (BLI) indicated that the affinity of the multivalent LaMs (LaM1-LaM8 and LaM8-LaM4) was higher than that of each LaM binding to mCherry. These BLI results suggested that using proper tandem multivalent nanobodies could enhance the binding affinity to mCherry. Although the k_on_ values of multivalent LaM1-LaM8 and LaM8-LaM4 binding to mCherry were similar to those of single nanobodies, their k_dis_ rates were significantly slower. The slower k_dis_ rates of the multivalent LaM1-LaM8 and LaM8-LaM4 led to an improved K_D_ compared to single nanobodies, highlighting the effectiveness of the tandem nanobody approach.

The BLI experiment for LaM8-LaM4 with DsRed was performed due to the structural similarity between DsRed and mCherry (Figure 5C). The results showed an exceptionally high affinity of LaM8-LaM4 to DsRed, which may contribute to LaM4′s binding (Appendix A).

The ITC results showed that the K_D_ of LaM1-LaM8 and LaM8-LaM4 to mCherry were 42.7 and 31.0 nM. (Figure 5D,E, Table 5). The binding affinity is higher than the LaM1 or LaM8 nanobody monomer (LaM1: 289 nM and LaM8: 199 nM). The number of binding sites (N) of LaM1-LaM8 and LaM8-LaM4 is lower than any of mono LaM, which may indicate that tandem nanobodies can bind more than one mCherry. For tandem dimer nanobodies, one nanobody includes two monomeric nanobodies. If both monomeric nanobodies bind to mCherry, the theoretical N value should be 1, whereas if only one monomeric nanobody binds, the N value is 0.5. Our experimental results showed that the N value of multimeric tandem nanobodies ranged between 0.5 and 1, indicating that some monomeric nanobodies were unable to bind to the corresponding epitopes on mCherry due to steric hindrance or other factors (including the formation of a ternary complex when a tandem nanobody simultaneously binds to two individual mCherry molecules instead of two epitopes on one mCherry molecule).

## 3. Discussion

### 3.1. Comparison of the Binding Mode of LaM Nanobodies to mCherry

It is uncommon to have binding information for six nanobody complexes targeting the same antigen. Therefore, we carefully analyzed and compared these mCherry-LaM complex structures. The buried surface area of the six nanobodies and mCherry are listed in the interface area column of Table 6. A positive correlation exists between buried surface area and affinity, where a more extensive interface area leads to higher affinity. However, the formation of special hydrogen bonds, salt bridges, and π–π stacks may also have a significant impact. The unusually high affinity of LaM3 may be attributed to the induction of a (mCherry-LaM3)_2_ heterotetramer formation.

### 3.2. Structural Insights of the Binding between DsRed-LaM3 and DsRed-LaM4

Tetrameric RFPs possess multiple chromophores, which allows for brighter fluorescence signals. This has led to their use in various applications, such as super-resolution microscopy, to achieve higher-resolution imaging of cellular structures. Investigating the molecular basis of the binding of certain LaM nanobodies to tetrameric DsRed may provide insight into the control of this protein and its potential applications as a tool in research. In our qualitative BLI binding assay, only LaM3 and LaM8-LaM4 were found to bind to DsRed (Appendix A), which is consistent with previous reports that only LaM3 and LaM4 bound to DsRed [36], while the other LaM nanobodies did not show binding to DsRed.

We thus analyzed the molecular mechanism of how LaM3 can bind to DsRed, as LaM2, LaM3, and LaM6 bind to similar epitopes on mCherry, but only LaM3 binds to DsRed (Figure 4D, Figure 6A and Appendix A). We first compared the amino acid sequences of mCherry and DsRed, particularly the binding amino acids, to the LaM nanobodies shown in the crystal structure (Appendix A). In DsRed, the amino acids corresponding to E158 and K167 in mCherry, which interact with LaM1, are replaced by R153 and H162, respectively (Appendix A). The amino acid corresponding to K187 in mCherry, which interacts with LaM2, is replaced by M182 in DsRed (Figure 6C and Appendix A). The residues recognized by LaM6 in mCherry, R130, T132, and T185, are replaced by I125, V127, and I180 in DsRed (Figure 6E and Appendix A), respectively. Additionally, the residue recognized by LaM8 and R169 in mCherry is replaced by A164 in DsRed (Appendix A). For LaM1, LaM2, LaM6, and LaM8, the residues recognized by them in mCherry have significant changes in both charge and steric hindrance compared to DsRed (Appendix A). Thus, they cannot bind to DsRed.

The molecular mechanism underlying the cross-reactivity of LaM3, which enables it to bind to both DsRed and mCherry, was analyzed in this study. Our results revealed that the amino acids responsible for the interaction between LaM3 and mCherry, namely S26, L129, and R130, are replaced by T21, F124, and I125, respectively, in DsRed Figure 6D and Appendix A. Notably, substituting R130 with I125 disrupts the ion–π interaction between mCherry and LaM3 (Figure 2F and Figure 6D), reducing the binding affinity of LaM3 to DsRed (Figure 4B,D). Although S26 and L129 in the first reported mCherry structure (corresponding to S21 and L124 in the mCherry-LaM3 structure in this study) participate in the interaction with LaM3 through the main chain N or O (Figure 2F), they are unlikely to affect the binding of LaM3, based on theoretical considerations.

The alignment of LaM3 with the DsRed tetramer indicates that other DsRed monomers in the tetramer are sterically excluded, suggesting that the observed binding of LaM3 to DsRed is likely to occur with a small population of DsRed monomers in solution. This binding could disrupt the formation of the DsRed tetramer, rendering LaM3 unsuitable as a tool for binding to DsRed, as its binding could alter the critical tetrameric state of DsRed.

The residues of mCherry interacting with LaM4 are consistent with the corresponding residues of DsRed (Appendix A), which should be the structural basis for LaM4 to bind DsRed and mCherry simultaneously. In short, through structural analysis, the difference in recognition of mCherry and DsRed of LaM nanobodies is mainly due to their recognition of specific residues.

Despite the advantages of using multimeric FPs that have been previously listed, there is a significant limitation in fluorescence imaging applications due to the potential for such multimeric FPs to alter the biological function of labeled target proteins significantly. Therefore, in most cases, monomeric FPs are strongly preferred over their multimeric counterparts.

### 3.3. Possible Applications of Multimeric Tandem Nanobodies

Multimeric tandem nanobodies have several advantages, such as the slow dissociation property (LaM8-LaM4 and LaM1-LaM8) demonstrated in this study. This property allows nanobodies to stably bind to proteins for extended periods and to bind to antigens even when the solvent is changed harshly. Therefore, compared to mono-nanobodies, tandem nanobodies are more suitable for in vitro binding experiments, capture assays, purification assays, etc. For example, in magnetic bead-coupled nanobody capture antigen assays, using tandem nanobodies can result in a higher purity of the target protein by allowing for multiple and longer washing steps to remove impurities while ensuring minimal loss in yield. We found that LaM1 and LaM8 in this study and LaM2 [38] reported previously might have better potential to capture the mCherry-fusion protein through epitopes on the flank of β-barrel of mCherry. However, not every tandem nanobody construct is successful. In addition to LaM1-LaM8 and LaM8-LaM4, we designed LaM6-LaM4. The same batch of LaM6-LaM4 was challenging to express in bacteria.

Although monomeric LaM4 has a very high affinity to mCherry among these LaM nanobodies [36,38], its recognizing epitope to mCherry makes it a bit difficult to bind mCherry-tagged chimera proteins in living cells, as the binding site of LaM4 is not located in the flank of the β-barrel but rather at the N- and C-termini on the top of the barrel. If the linker connecting LaM4 to another protein in a chimera construct is not suitable, steric hindrance can occur (Figure 6A), and this was proven by a trapping experiment, which resulted in low yield compared to other LaM nanobodies [37].

The binding epitopes of LaM8 are different from others, while the epitope of LaM3 is similar to LaM2 and LaM6 (Figure 6A). The alignment results also clearly showed that there is a steric hindrance from residues between LaM1 and LaM2 (Figure 6F), LaM3 (Figure 6G), and LaM6 (Figure 6H).

Despite both monomeric proteins, we also observed an interesting phenomenon of induced hetero-tetramerization between LaM3 and mCherry. The formation of the (mCherry-LaM3)_2_ heterotetramer was observed in the crystal structure, where specific interactions between the nanobody and mCherry Lys15 and Glu28 were observed (Figure 3A,B,E,F, Figure 6B and Appendix A).

Non-covalent binding has a minor impact on the conformation of antigens and antigen-fused proteins than covalent binding. This may help to maintain the physiological activity of the antigen. Tandem nanobodies also can target different antigens, resulting in the spatial proximity of foreign antigens under physiological conditions. This enables the observation of downstream biological reactions that occur after multiple antigen targets are brought into proximity. This approach is valuable for studying protein-protein interactions. It can even induce proximity between other biomolecules, such as DNA/RNA and proteins, allowing for investigating their regulatory roles in transcription and translation.

## 4. Materials and Methods

### 4.1. Protein Expression

The protein sequences of the mCherry nanobodies LaM1, LaM3, and LaM8 were the same as Fridy et al. described (Appendix A) [36]. The codon sequences were optimized based on favored codon usage in *Escherichia coli*, synthesized by Generay (Shanghai, China), and inserted into plasmid pET28a-SUMO used previously with BamHI and XhoI endonuclease cleavage sites [38]. These vectors expressing fusion proteins were transformed into BL21(DE3) *E. coli*. The cells were grown in LB medium at 37 °C (220 rpm) until OD600 to ~0.6, precooled to 20 °C for about 30 min, then induced at 20 °C with 0.3 mM IPTG about 20 h. The bacteria were collected by centrifuging twice (5000 rpm, 15 min, 4 °C) and frozen at −20 °C for further purification.

The plasmid of mCherry was from Tsien Lab (Appendix A) [15] and further subcloned to a pET21a-derived vector, which contains a 10 × His tag and a TEV protease site at the N terminal of mCherry. The procedure of mCherry’s expression is the same as the LaM nanobodies.

### 4.2. Protein Purification

Frozen cells were thawed and re-suspended in 40 mL of lysis buffer containing NiA (50 mM Tris-HCl [pH 9.0], 150 mM NaCl, 20 mM imidazole, 5% glycerol), 0.2 mM TCEP, and 0.02% (*v*/*v*) Triton X-100. Cells were lysed by a high-pressure cell homogenizer (ATS, Suzhou, China) at 4 °C. The lysate was centrifuged at 17,000× *g* for 50 min at 4 °C. The supernatant was gently mixed with 10 mL Ni-NTA resin (Cytiva, Marlborough, MA, USA) in a gravity column for about 30 min at room temperature, then equilibrated by 50 mL NiA, washed with 30 mL NiC (50 mM Tris-HCl [pH 9.0], 1 M NaCl, 20 mM imidazole, 5% glycerol), and equilibrated by 50 mL NiA. Half of the protein on the resin was eluted by NiB (50 mM Tris-HCl [pH 9.0], 150 mM NaCl, 300 mM imidazole, 5% glycerol) and flash-frozen in liquid nitrogen, then stored at -80 °C for most binding assay experiments. The other half of the protein was cleaved by ULP1 (for His-SUMO-LaM1, LaM3 & LaM8) or TEV protease (for H10T-mCherry) overnight (about 12–16 h), then eluted by NiA with 0.5 mM TCEP. We also cleaved SUMO-LaM by ULP1 in NiB buffer over 2 h on ice, exchanged to QA buffer (50 mM Tris-HCl [pH 8.0], 0.5 mM TCEP) with a 5 mL desalting column and purified protein with two series-connected 5 mL prepacked Ni-NTA column to obtain purified LaM nanobodies for further structural related experiments. The proteins were flash-frozen in liquid nitrogen and then stored at −80 °C. After measuring the absorption of proteins at 280 nm, we gently mixed mCherry with the nanobodies (LaM1, LaM3, or LaM8) with a molar ratio of 1: 1.25 to ensure the mCherry is sufficiently bonded with each nanobody. The complex of mCherry and its nanobody for crystallization were concentrated using a 10 kD ultra filtrate column (Amicon, Miami, FL, USA) by centrifuge (4000× *g*, 4 °C) until the concentration was up to 10–20 mg/mL. The excess nanobody was separated by Superdex 200 Increase 10/300 GL gel filtration column (Cytiva, Marlborough, MA, USA) with a buffer containing 20 mM HEPES (pH 8.0) and 150 mM NaCl. Fraction samples were added 0.2 mM TCEP and mixed gently, flash-frozen with liquid nitrogen, and stored at −80 °C for crystallization. All procedures were processed on ice.

### 4.3. Crystallization and Structural Determination of mCherry-LaM1, mCherry-LaM3, and mCherry-LaM8

To obtain the crystal of the mCherry-nanobody complexes, a frozen purified complex of mCherry and respective nanobody were thawed on ice and concentrated to about 20–30 mg/mL using Amicon Ultra 0.5 mL Centrifugal Filters (Merck Millipore, Darmstadt, Germany) by centrifuge at 4000× *g*, 4 °C. Complexes were crystallized in sitting diffusion by mixing 200 nL protein solution with 200 nL precipitant with Gryphon (Art Robbins, Sunnyvale, CA, USA). The crystal of mCherry-LaM1 was obtained in 0.2 M potassium sulfate, 20% (*w*/*v*) PEG 3350; the crystal of mCherry-LaM3 was obtained in 0.2 M sodium sulfate, 0.1 M Bis-Tris propane (pH 7.5), 20% (*w*/*v*) PEG 3350; and the crystal of mCherry-LaM8 was obtained in 0.2 M magnesium chloride, 0.1 M Tris-HCl (pH 8.5), 30% (*w*/*v*) PEG 4000.

X-ray diffraction data were collected at beamlines BL17U1, BL19U1, and BL10U2 of the Shanghai Synchrotron Radiation Facility [43]. The diffraction data were processed by HKL3000 software (Release 720) [44]. The structures of three mCherry-LaMs complexes were obtained by molecular replacement using PHENIX (Version 1.20.1) [45] and CCP4i (Version 8.0) [46], while predicted nanobody models from Phyre2 (accessed 15/11/2022) [47] and mCherry (PDB ID: 2H5Q) from Protein Data Bank were used as search models in molecular replacement. Refinement was performed with Refmac5 in CCP4i and PHENIX, and adjustments were made with COOT (Version 0.8.9.1) [48]. The corresponding figures were drawn by Pymol (Version 2.6) [49].

### 4.4. Verification of Related Protein or Protein Complex by Mass Spectrometry

Samples from every step were loaded into 4–20% SDS-PAGE (Tanon, Shanghai, China) to determine the approximate molecular weight. To determine the precise molecular weight of the protein sample, the linear mode matrix-assisted laser desorption ionization coupled to time of flight mass spectrum (MALDI-TOF/MS) was performed with a FLEX MALDI-TOF instrument (Bruker Instruments, Inc., Billerica, MA, USA) after that the buffer of the sample was exchanged to 10 mM NH_4_Ac by a Superdex 200 Increase 5/150 GL (Cytiva, Marlborough, MA, USA) column.

### 4.5. Fluorescence Detected Size Exclusion Chromatography (FSEC)

To determine whether LaM nanobodies binds to mCherry and if there is competitiveness among LaM nanobodies to the mCherry, we set different experiment groups to contain several LaM nanobodies and mCherry as experiment samples [41]. LaM nanobodies and mCherry were added with a final molar ratio of 4:3. The concentration of LaM nanobody to mCherry is 10 μM to 7.5 μM, in FSEC buffer (20 mM HEPES, pH 7.4; 150 mM NaCl; 0.2 μm-diameter membranes filtrated), then mixed gently and incubated more than 1 h at room temperature. After high-speed centrifugation (20,000× *g*, 20 min, 4 °C), 50 μL of supernatants were loaded onto a Superdex 200 Increase 10/300 GL (Cytiva, Marlborough, MA, USA), equilibrated with FSEC buffer. Additionally, mCherry without LaM was set as the control group. The fluorescence of each sample was recorded by a fluorometer (excitation, 587 nm; emission, 610 nm). The data were processed and analyzed by OriginPro software 2021 [50].

### 4.6. Affinity and Kinetic Parameter Determination by BLI (OctetRed96)

Nitrilotriacetic acid (NTA) capture tips (ForteBio) were soaked in 1 × kinetics buffer (20 mM HEPES [pH 7.4], 150 mM NaCl, 0.005% [*v*/*v*] Tween-20) for 10 min, and a baseline signal was obtained over the 60 s. Individual his-tagged SUMO-LaMs were immobilized on the biosensor tips until the thickness was up to 2 nm. After a wash step in 1× kinetics buffer for the 60 s, the association of mCherry was determined at different gradient concentrations over 10 min. The dissociation in 1 × kinetics buffer was measured over 10 min. Ka, kd, and K_D_ values were calculated using a 1:1 global fit model for mono nanobody and a 2:1 heterogeneous ligand binding model for tandem nanobody to antigen with ForteBio 10.0 data analysis software (Version 10.0.0.5).

### 4.7. Isothermal Titration Calorimetry (ITC)

To determine the proper thermal parameters, we performed an ITC assay using the iTC200 instrument kept at 25 °C or 20 °C. For each experiment, about 70 μL of 100 μM syringe sample was injected into the cell in which there are about 300 μL of 10 μM cell sample. 0.4 μL for drop 1 and 1.5 μL for drop 2–25 is used in most experiments. 0.4 μL for drop 1, 3 μL for drop 2–5, 1 μL for drop 6–9, 0.3 μL for drop 10–25, 1 μL for drop 26–29, 3 μL for drop 30–33 was processed to obtain more experiment data near the halfway point of the curve, and this was designed with MicroCal PEAQ-ITC Analysis Software based on the data of Kd of LaM3 and mCherry [36]. Data were analyzed with MicroCal PEAQ-ITC Analysis Software (Version 1.0), and one set of sites was used as the fitting model.

### 4.8. Fluorescence Spectroscopy

The emission spectra of mCherry with LaMs were measured while the excitation wavelength was fixed at 560 nm. Additionally, the excitation spectra were measured while the emission wavelength was set at 610 nm. The pure mCherry protein without LaM was used as the control group. The final concentration of mCherry in all groups was equal, and SUMO-LaMs were added as the molar ratio of LaM to mCherry as 1.25:1, and then the sample was diluted with GF buffer. The spectra were obtained with a fluorescence spectrophotometer (Hitachi, Japan). The emission spectra were acquired between 580 nm and 700 nm. The spectra were analyzed with OriginPro software 2021.

## 5. Conclusions

We determined the structures of the mCherry complex with LaM1, LaM3, and LaM8 nanobodies, which had yet to be previously resolved. We systematically analyzed the interactions between these nanobodies and the red fluorescent proteins, and based on this analysis, we designed potentially useful tandem multivalent nanobodies with specific binding to mCherry.

## Figures and Tables

**Figure 1 ijms-24-06952-f001:**
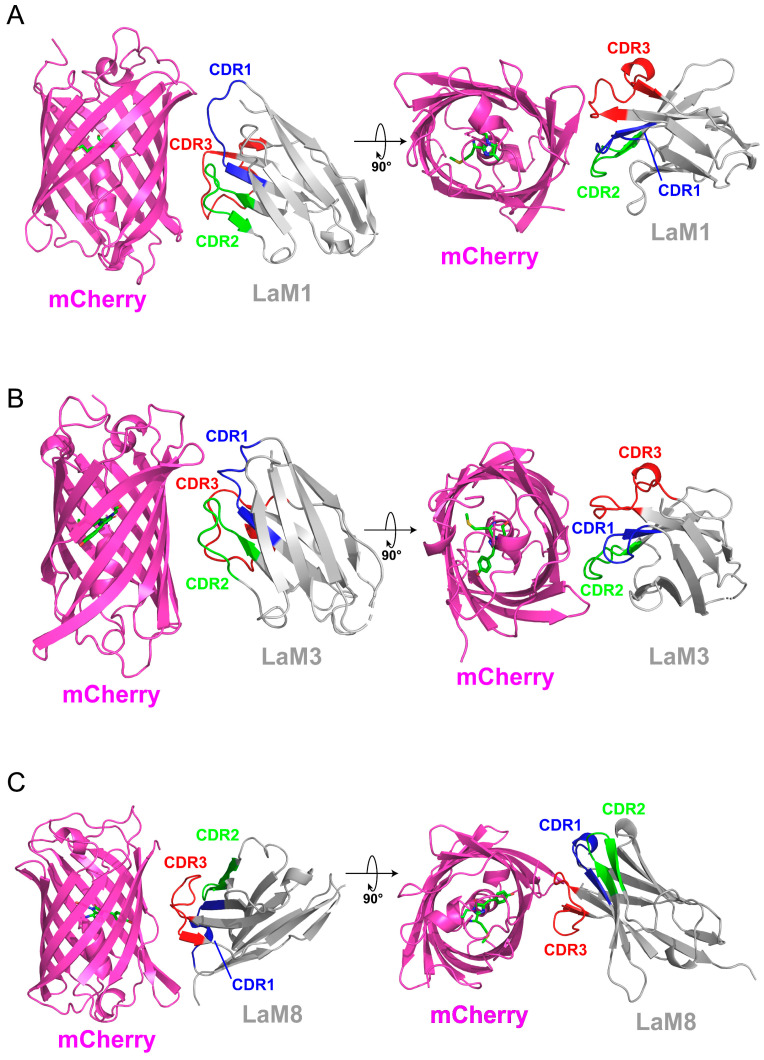
Overall crystal structures of mCherry-LaM complexes. CDR1, CDR2, and CDR3 are shown in blue, green, and red, respectively, and non-hypervariable regions are gray, mCherry is magenta, and the chromophore is a stick-like structure in green. (**A**) A schematic representation of the mCherry-LaM1 heterodimer. (**B**) A schematic representation of the mCherry-LaM3 heterodimer. (**C**) A schematic representation of the mCherry-LaM8 heterodimer.

**Figure 2 ijms-24-06952-f002:**
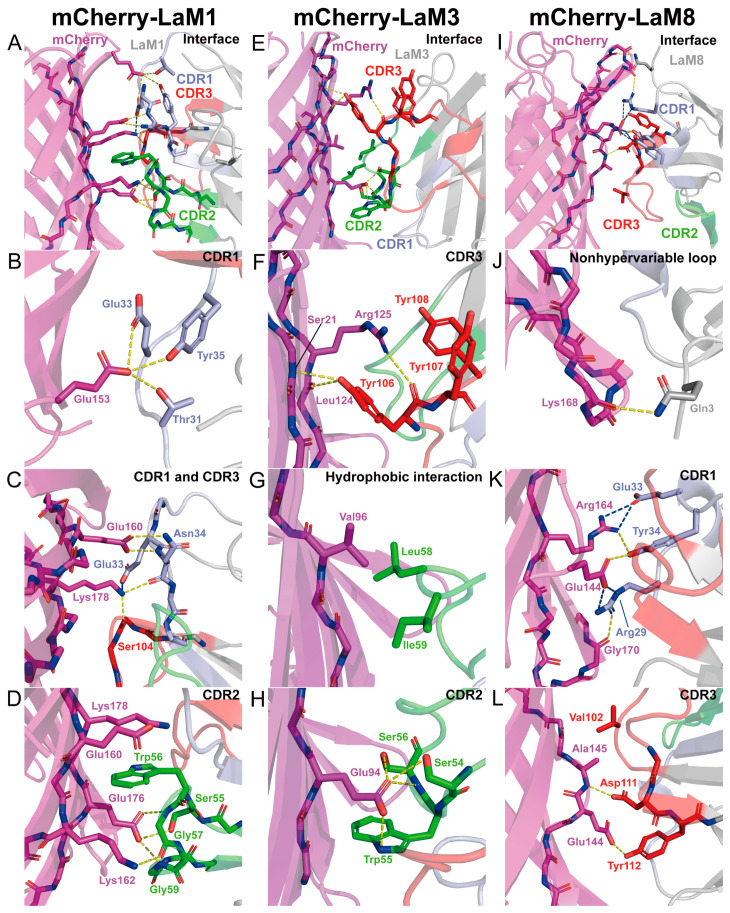
Interaction surface between mCherry and LaM nanobodies in the crystal structure of mCherry-LaM nanobody complexes. (**A**,**E**,**I**) show overviews of the interface for mCherry-LaM1, mCherry-LaM3, and mCherry-LaM8, respectively. The mCherry is colored pink and LaM nanobodies in grey, the CDR1, CDR2, and CDR3 of LaM nanobodies are colored in light blue, green, and red, respectively. Panels (**B**–**D**), (**F**–**H**), and (**J**–**L**) show detailed interactions between LaM1 and mCherry, LaM3 and mCherry, and LaM8 and mCherry, respectively. Hydrogen bonds are shown in yellow dashes, salt bonds are shown in blue dashes, and key residues are depicted as sticks.

**Figure 3 ijms-24-06952-f003:**
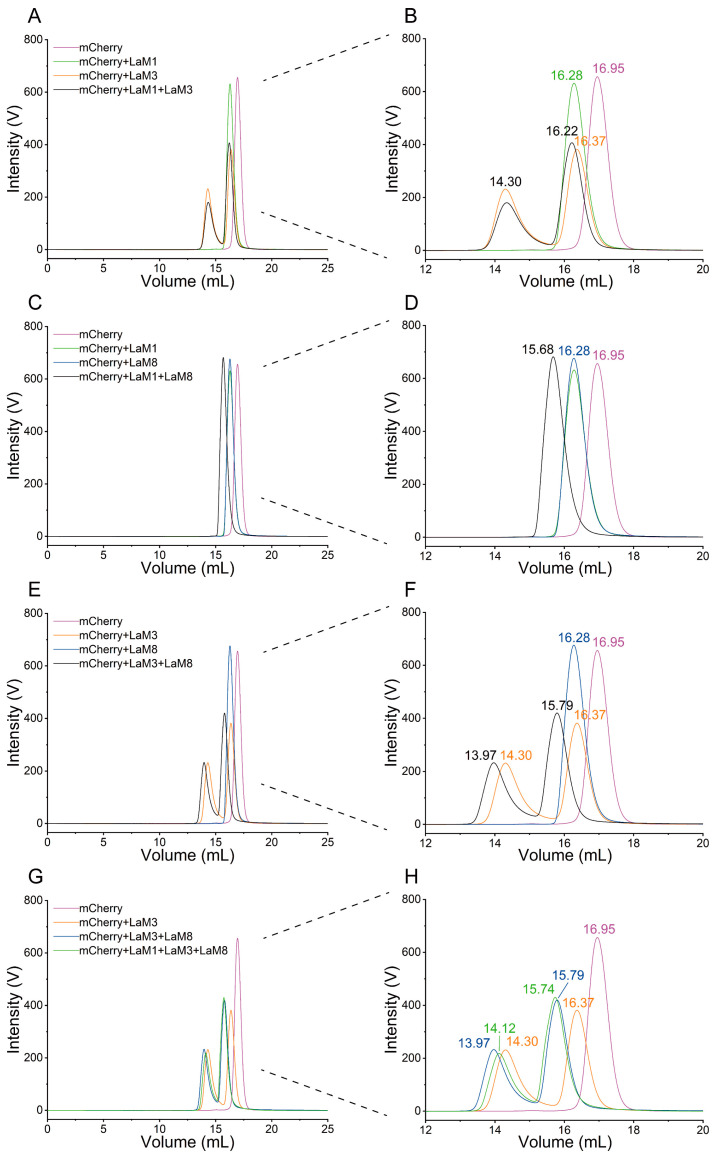
Verify the binding of mCherry to LaM nanobodies by fluorescence-detected size exclusion chromatography. The *x*-axis represents elution volume, and the *y*-axis represents the detected fluorescence intensity by the fluorescence detector. Panels (**A**,**C**,**E**), and (**G**) show the FSEC profiles of mCherry with or without LaMs, while panels (**B**,**D**,**F**), and (**H**) zoom in on the details of the 12–20 mL range for the corresponding FSEC sub-figures. Panels (**A**,**B**) compare the elution volume of mCherry with that of mCherry-LaM1, mCherry-LaM3, and the mCherry-LaM1/LaM3 complex; panels (**C**,**D**) compare the elution volume of mCherry with that of mCherry-LaM1, mCherry-LaM8, and the mCherry-LaM1/LaM8 complex; panels (**E**,**F**) compare the elution volume of mCherry with that of mCherry-LaM3, mCherry-LaM8, and the mCherry-LaM3/LaM8 complex; panels (**G**,**H**) compare the elution volume of mCherry with that of mCherry-LaM3, the mCherry-LaM3/LaM8 complex, and the mCherry-LaM1/LaM3/LaM8 complex.

**Figure 4 ijms-24-06952-f004:**
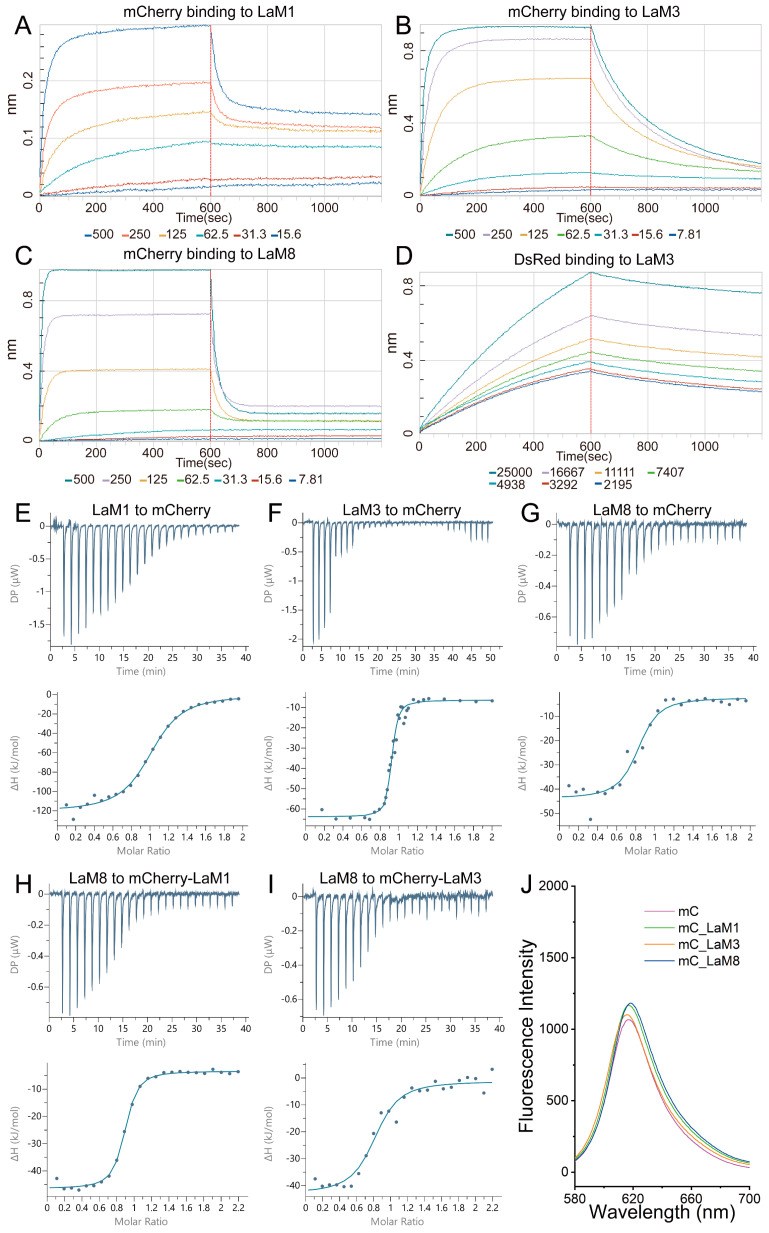
Characterize the binding kinetics and thermodynamic and fluorescence parameters of the binding between mCherry and LaM nanobodies (**A**–**D**) Binding kinetics of mCherry or DsRed with LaM nanobodies by BLI. Sumo−His tagged proteins were loaded onto Ni−NTA biosensors, then equilibrated before setting the baseline to zero at t = 0, and the association within mCherry or DsRed is in the range of 0−600 s. The dissociation in kinetics buffer is in the range of 600−1200 s. The *x*−axis represents time, and the *y*−axis represents the amount of protein binding to the biosensor. (**A**–**C**) A total of 100 nM His−SUMO−LaM1, His−SUMO−LaM3, and His−SUMO−LaM8 were loaded onto sensors, respectively, and 500, 250, 125, 62.5, 31.3, 15.6, and 0 nM mCherry tested for LaM1; 500, 250, 125, 62.5, 31.3, 15.6, 7.81, and 0 nM mCherry tested for LaM3 and LaM8 were assayed for association and dissociation. (**D**) A total of 100 nM His−SUMO−LaM3 were loaded onto sensors, and 25.4, 16.7, 11.1, 7.41, 4.94, 3.29, 2.20, and 0 μM DsRed were assayed for association and dissociation. (**E**–**I**) Binding affinity of mCherry with LaM nanobodies by ITC. Each subfigure shows raw data (upper) and a fitted curve (lower). The *x*−axis of the upper panels represents time, and the *y*−axis represents differential power (DP) between the sample cell and the reference cell. The *x*−axis of the lower panels represents the molar ratio, and the *y*−axis represents the molar change of calories in the syringe. Dots represent the acquired data in experiments, and lines represent the fitted curve. (**E**–**G**) Result of ITC of 100 μM LaM1, LaM3, LaM8, respectively, as syringe dropped into 10 μM mCherry in the cell. (**H**,**I**) Result of ITC of 100 μM LaM8 as syringe dropped into 8.9 μM mCherry−LaM1 or mCherry−LaM3 in cell. (**J**) The emission fluorescence spectra of mCherry when binding with LaM nanobodies. The mCherry protein or mCherry−LaM complexes were excited by a laser at 560 nm, and the fluorescence intensity was detected in the wavelength range from 580 to 700 nm.

**Figure 5 ijms-24-06952-f005:**
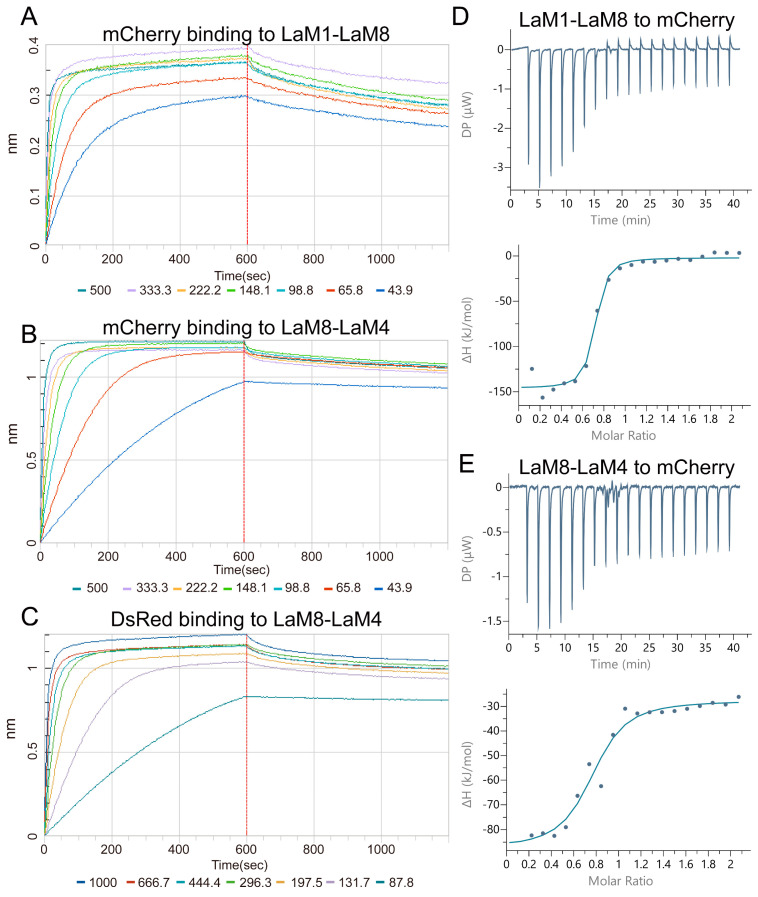
The binding kinetics and thermodynamics of multivalent LaM nanobodies to red fluorescent proteins (**A**–**C**) Binding kinetics of mCherry or DsRed with tandem LaMs by BLI. His−Sumo−LaM1−LaM8 and His−Sumo−LaM8−LaM4 proteins were loaded onto Ni−NTA biosensors, then equilibrated before setting the baseline to zero at t = 0, and the association within mCherry or DsRed is in the range of 0–600 s. The dissociation in kinetics buffer is in the range of 600–1200 s. The fitting model is 2:1 (heterogeneous ligand). The *x*−axis represents time, and the *y*−axis represents the amount of protein binding to the biosensor. (**A**,**B**) A total of 1000 nM His−SUMO−LaM1−LaM8 and His−SUMO−LaM8−LaM4 were loaded onto sensors, 500, 333.3, 222.2, 148.1, 96.8, 65.8, 43.9, and 0 nM mCherry were assayed for association and dissociation. (**C**) A total of 100 nM His−SUMO−LaM8−LaM4 was loaded onto sensors, and 1000, 666.7, 444.4, 296.3, 197.5, 131.7, 87.8, and 0 nM DsRed were assayed for association and dissociation. (**D**,**E**) ITC of mCherry with tandem multivalent LaMs. Each subfigure shows raw data (upper) and a fitted curve (lower). Dots represent the acquired data in experiments, and lines represent the fitted curve. A total of 100 μM LaM1−LaM8 (**D**) or LaM8−LaM4 (**E**) as syringe dropped into 10 μM mCherry in the cell. The *x*−axis of the upper panels represents time, and the *y*−axis represents differential power (DP) between the sample cell and the reference cell. The *x*−axis of the lower panels represents the molar ratio, and the *y*−axis represents the molar change of calories in the syringe.

**Figure 6 ijms-24-06952-f006:**
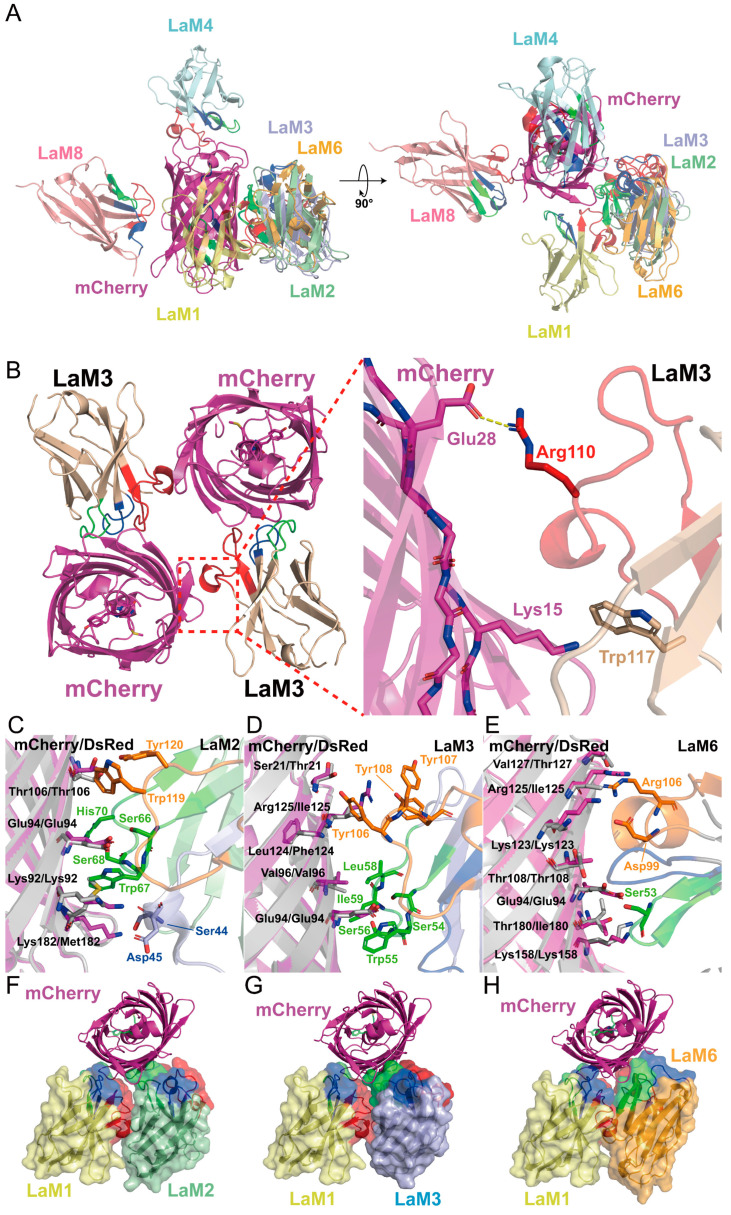
(**A**) Aligned structures of mCherry with LaM nanobodies. Structures of complexes from PDB and in this article were aligned to mCherry (PDB ID: 2H5Q) by Pymol. The mCherry is shown as magenta, LaM1 is pale yellow, LaM2 (PDB ID: 6IR2) is pale green, LaM3 is light blue, LaM4 (PDB ID: 6IR1) is pale cyan, LaM6 (PDB ID: 7SAL) is bright orange, and LaM8 is salmon. CDR1 is in blue, CDR2 is in green, and CDR3 is in red. (**B**) Detailed interaction of tetramer of (mCherry-LaM3)_2_. The left subfigure is an overview of the tetramer of (mCherry-LaM3)_2_, and the right subfigure is the interaction between LaM3 and mCherry, forming a tetramer. Of LaM3, CDR3 is shown in red, non-hypervariable region is shown in light orange. Key residues are shown as sticks, in which atom N is in blue and atom O is in red. (**C**,**E**) Interaction between LaM2, LaM3, or LaM6 binding to mCherry or DsRed. DsRed (PDB ID: 1ZGO), a complex of mCherry-LaM2, mCherry-LaM3, and mCherry-LaM6, was aligned to mCherry. CDR1 is shown in light blue (**C**) or blue (**D**,**E**), CDR2 is green, CDR3 is orange, mCherry is magenta, and DsRed is silver. Key residues involving interaction are shown as sticks. (**F**–**H**) Structures of mCherry-LaM1 aligned to mCherry-LaM2, mCherry-LaM3, and mCherry-LaM6. The surface of LaMs is shown transparently, mCherry is in magenta, LaM2 in pale green, LaM3 in light blue, and LaM6 in bright orange. CDR1 in blue, CDR2 in green, and CDR3 in red.

**Table 1 ijms-24-06952-t001:** Data collection and refinement statistics. The values in parentheses are for the high-resolution shells.

	mCherry-LaM1	mCherry-LaM3	mCherry-Lam8
Wavelength	0.979	0.979	0.979
Resolution range	31.24−2.05 (2.123−2.05)	66.8−3.29 (3.408−3.29)	50.36−1.31 (1.357−1.31)
Space group	P 1	C 1 2 1	P 2_1_ 2_1_ 2_1_
Unit cell	68.07 74.83 94.64 111.33 100.68 104.42	134.867 104.831 68.5085 90 102.828 90	63.0748 75.3103 83.6246 90 90 90
Total reflections	349,349 (33,627)	93,208 (7800)	1,055,598 (41,572)
Unique reflections	97,578 (9741)	14,164 (1388)	95,041 (8553)
Multiplicity	3.6 (3.5)	6.6 (5.6)	11.1 (4.9)
Completeness (%)	96.90 (96.43)	99.43 (99.64)	98.81 (90.40)
Mean I/sigma(I)	8.97 (1.90)	4.67 (1.59)	18.48 (1.71)
Wilson B-factor	41.13	65.43	14.25
R-merge	0.08522 (1.024)	0.4263 (1.557)	0.06234 (0.7762)
R-meas	0.1008 (1.218)	0.4681 (1.733)	0.06521 (0.8674)
R-pim	0.05308 (0.6503)	0.19 (0.7503)	0.01877 (0.3708)
CC1/2	0.996 (0.565)	0.835 (0.472)	0.999 (0.645)
CC*	0.999 (0.85)	0.954 (0.801)	1 (0.886)
Reflections used in refinement	97,558 (9740)	14,108 (1383)	95,021 (8548)
Reflections used for R-free	4722 (448)	682 (90)	2000 (180)
R-work	0.2043 (0.3369)	0.2638 (0.3527)	0.1780 (0.2628)
R-free	0.2350 (0.3622)	0.3123 (0.4489)	0.1896 (0.2445)
CC (work)	0.961 (0.806)	0.897 (0.810)	0.967 (0.803)
CC (free)	0.946 (0.716)	0.811 (0.638)	0.971 (0.763)
Number of non-hydrogen atoms	11,069	5076	3231
macromolecules	10,428	5030	2621
ligands	114	46	23
solvent	527	0	587
Protein residues	1353	671	338
RMS (bonds)	0.002	0.002	0.006
RMS (angles)	0.53	0.54	0.9
Ramachandran favored (%)	98.42	95.27	98.8
Ramachandran allowed (%)	1.51	3.97	1.2
Ramachandran outliers (%)	0.08	0.76	0
Rotamer outliers (%)	0.95	0.41	0
Clashscore	4.34	5.03	3.67
Average B-factor	48.56	74.35	19.15
macromolecules	48.44	74.38	16.2
ligands	53.5	71.52	19.52
solvent	49.87		32.29

**Table 2 ijms-24-06952-t002:** The binding kinetics and affinity parameters of LaM1, LaM3, and LaM8 nanobodies to mCherry or DsRed by bio-layer interferometry.

Antibody	Antigen	K_D_ (M)	k_on_ (1/Ms)	k_dis_ (1/s)	R^2^
LaM1	mCherry	1.29 × 10^−8^ ±2.80 × 10^−10^	9.13 × 10^4^ ±1.76 × 10^3^	1.17 × 10^−3^ ±1.19 × 10^−5^	0.9426
LaM3	mCherry	3.06 × 10^−8^ ±2.94 × 10^−10^	1.06 × 10^5^ ±9.72 × 10^2^	3.24 × 10^−3^ ±9.63 × 10^−6^	0.9896
LaM8	mCherry	3.68 × 10^−8^ ±1.50 × 10^−9^	2.52 × 10^5^ ±9.88 × 10^3^	9.26 × 10^−3^ ±1.03 × 10^−4^	0.9354
LaM3	DsRed	7.13 × 10^−6^ ±1.31 × 10^−7^	62.2 ±1.02	4.43 × 10^−4^ ±3.56 × 10^−6^	0.9838

**Table 3 ijms-24-06952-t003:** The binding parameters and relative thermodynamic parameters between LaM nanobodies to mCherry or mCherry-LaM complex by isothermal titration calorimetry.

Group	Temp (°C)	N (Sites)	K_D_ (M)	∆H (kJ/mol)	∆G (kJ/mol)	−T∆S (kJ/mol)	ΔS (J/[mol × K])
LaM1 to mCherry	25	1.000 ±1.50 × 10^−2^	2.89 × 10^−7^ ±5.18 × 10^−8^	−120 ±3.79	−37.4	82.9	−278.19
LaM3 to mCherry	25.1	0.913 ±9.50 × 10^−3^	1.37 × 10^−8^ ±5.21 × 10^−9^	−57.5 ±0.657	−44.9	12.5	−41.93
LaM8 to mCherry	25.2	0.842 ±4.80 × 10^−2^	1.99 × 10^−7^ ±1.34 × 10^−7^	−46.8 ±6.32	−38.3	8.51	−28.54
LaM8 to mCherry with LaM1	25.1	0.854 ±5.40 × 10^−3^	5.40 × 10^−8^ ±7.68 × 10^−9^	−43 ±0.569	−41.5	1.51	−5.07
LaM8 to mCherry with LaM3	25.1	0.799 ±3.00 × 10^−2^	2.36 × 10^−7^ ±1.03 × 10^−7^	−41.8 ±2.85	−37.9	3.95	−13.25

**Table 4 ijms-24-06952-t004:** The binding kinetics and affinity parameter of the multivalent nanobodies to mCherry or DsRed by bio-layer interferometry.

Antibody.	Antigen	K_D_ (M)	K_D2_ (M)	k_on_ (1/Ms)	k_on2_ (1/Ms)	k_dis1_ (1/s)	k_dis2_ (1/s)	R^2^
LaM1-LaM8	mCherry	1.87 × 10^−9^	<1.0 × 10^−12^	2.96 × 10^5^	4.67 × 10^4^	5.52 × 10^−4^	<1.0 × 10^−7^	0.986
	error	<1.0 × 10^−12^	4.52 × 10^−10^	2.10 × 10^3^	1.69 × 10^3^	4.63 × 10^−6^	−	
LaM8-LaM4	mCherry	1.38 × 10^−9^	1.29 × 10^−9^	1.73 × 10^5^	5.06 × 10^4^	2.39 × 10^−4^	6.51 × 10^−5^	0.990
	error	<1.0 × 10^−12^	1.01 × 10^−10^	1.12 × 10^3^	1.93 × 10^3^	1.89 × 10^−6^	4.47 × 10^−6^	
LaM8-LaM4	DsRed	2.64 × 10^−9^	4.12 × 10^−9^	8.81 × 10^4^	1.99 × 10^4^	2.32 × 10^−4^	8.20 × 10^−5^	0.991
	error	<1.0 × 10^−12^	3.12 × 10^−10^	5.05 × 10^2^	9.92 × 10^2^	1.71 × 10^−6^	4.69 × 10^−6^	

The fitting model is a 2:1 heterogeneous ligand. The molar concentration of DsRed (tetramer) is calculated as a monomer.

**Table 5 ijms-24-06952-t005:** The binding thermodynamic parameters between the multivalent nanobodies to mCherry by isothermal titration calorimetry.

Nanobody	Temp (°C)	N (Sites)	K_D_ (M)	∆H (kJ/mol)	∆G (kJ/mol)	−T∆S (kJ/mol)	ΔS (J/[mol × K])
LaM1-LaM8	20.1	0.671 ±1.20 × 10^−2^	4.27 × 10^−8^ ±1.82 × 10^−8^	−144 ±5.00	−41.4	103	−351.42
LaM8-LaM4	20.2	0.750 ±3.70 × 10^−2^	3.10 × 10^−7^ ±1.70 × 10^−7^	−60.1 ±6.09	−36.6	23.5	−80.15

**Table 6 ijms-24-06952-t006:** Analysis of protein buried surface in the complex by PISA on PDBe.

Protein 1	N_at_	N_res_	Surface (Å)	Protein 2	N_at_	N_res_	Surface (Å)	Interface Area (Å^2^)	ΔG (kcal/mol)	ΔG (*p*-Value)	N_HB_	N_SB_	N_DS_	CSS
LaM1 (8IM1)	64	18	6299	mCherry	75	20	9663	627.0	−1.2	0.494	13	3	0	0.000
LaM2 (6IR2)	71	17	6365	mCherry	90	22	9926	679.6	−5.1	0.160	9	1	0	0.000
LaM3 (8ILX)	60	16	6108	mCherry	67	20	9257	597.2	−8.6	0.235	6	0	0	0.100
LaM4 (6IR1)	76	23	6127	mCherry	72	21	10,088	757.8	−7.0	0.172	6	0	0	0.100
LaM6 (7SAL)	59	15	6268	mCherry	77	21	10,081	611.0	−5.2	0.257	4	1	0	0.000
LaM8 (8IM0)	66	20	6295	mCherry	75	23	9879	688.8	1.5	0.734	11	4	0	0.000

## Data Availability

All data are provided in the article and the Appendix A.

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
