# Peer review of "Structural Insights into the Binding of Red Fluorescent Protein mCherry-Specific Nanobodies"

_ijms, 2023, doi:10.3390/ijms24086952_

Round 1

Reviewer 1 Report

This paper by Liang et al is a follow-up of previous work (ref 38), in which the authors address the binding mechanism of nanobodies to the red fluorescent protein mCherry by x-ray crystallography and various photophysical techniques.

The work is generally interesting and well conducted, and I would recommend publication upon a few minor revisions listed below:

The English of the manuscript could be significantly improved. I would recommend the manuscript be carefully read and amended by a native English speaking collaborator. Some sentences make almost no sense due to bad English … As an example, look at the 1st sentence on the top of page 9 starting with “According to the fluorescence curve off mCherry …”

Abstract and other places in manuscript: RFP should be considered as a family of red fluorescent proteins, so always cited using plural (RFPs).

Quite many links for references were missing in the provided PDF manuscript (e.g. on page 3). Please correct this.

The crystallographic data on LAM3 look very weak (poor resolution and horrible R-merge). This might be related to the observed dimer formation ? The authors should extensively comment on this. In this condition, how reliable is the detailed interaction between mCherry and LAM3 shown in figure 2 and figure 6 ? Some words of caution should probably be added ?

X/Y axes and font size in figure 4 and 5 are too small, so very difficult to read, please improve. Please give all meanings of Y axes, for example what is DP ?

The very substantial discrepancy between the results of BLI and ITC is striking. The authors suggest that BLI results may be inaccurate (end of P9). Can they further elaborate on this ? Do they think that at least relative values measured by BLI between the different LAMs remain trustable ? Based on ITC, it looks like the Kd of the various nanobodies are quite high (10-100 nM). How does this compare to known Kd in the case of e.g. anti-GFP nanobodies and nanobodies in general ? Please comment.

Table 5: it is unclear to this reviewer what the N<1 mean in this case ? could this point at the formation of a population of dimeric mCherry, with each monomer binding a single nanobody ? Please comment.

P12 L361-365: confusion between nm and nM ? “maximum thickness” ??

 P14 L393: “If both nanobodies in the tandem construct bind to a single mCherry, the theoretical N value would be 0.5”. Same issue as above … Maybe I don’t get it, but should not this be phrased as “If both nanobodies in the tandem construct bind to a different mCherry, the theoretical N value would be 0.5” ?

P14 paragraph 3.2: the advantages of multimeric FPs listed by the authors are also offset with the big caveat in fluorescence imaging applications that labeling with such multimeric FPs can drastically alter the biological function of the labeled target proteins. So in most applications, monomeric FPs are much preferred. This should be added to the discussion.

In their discussion, could the authors speculate more precisely about why the LAMs nanobodies do not alter the fluorescence behavior of mCherry, as opposed to what is generally observed with GFP nanobodies (GFP-enhancer for example, ref 22 )

Reviewer 2 Report

This manuscript described a comprehensive work on the interactions between RFP and its corresponding nanobodies. The authors carefully analyzed the structures and binding kinetics of the complexes and provided very important insights into the further engineering of the nanobody. Although the work was done in a rational design and was performed with good quality, the writing and figures need sufficient revision. A Systematic text editing by a native speaker or a professional company is strongly suggested.  Some of the figures are labelled inappropriately (see detail comments below). Some specific issues are listed below.

Page 3, the “2.1. Protein expression, purification, and characterization” section is a simple repeat of the method section and thus can be removed or can be combined to the method section.

Page 3, some references are missing. 

Page 3, “by TEV/ULP1 protease on the column’ should be “by TEV and ULP1 protease on the column, respectively”.

Fig 2. The distances of hydrogen bonds are labelled too small and is not necessary. Please remove the distances. The residues labelling should be enlarged. Three letter coding cannot be all capitalized. For example, it should be Glu33 but not GLU33. Do not put the labelling onto the ribbons, which makes it hard to see.

Fig 6C. Too many red and magenta ribbon and labelings. I can see nothing. Please make it much simpler and easy to follow

Page 16 “Why can LaM3 still bind to DsRed? . It is quite rare to describe results with a question. Please reword.

Reviewer 3 Report

The authors expressed, crystallized and solved the structures of complexes of RFP mCherry with its specific nanobodies LaM1, LaM3, and LaM8. The authors characterized properties including structural information, binding epitope, affinity, thermodynamic and kinetic properties of these 3 complexes using various technologies. Based on these data and published data on other RFP nanobodies, the authors designed multivalent tandem nanobodies LaM1-LaM8 and LaM8-LaM4, and found that they had higher affinity and specificity to mCherry. This research presents new information on how nanobodies can be used to target specific proteins, which could lead to the development of improved tools for manipulating mCherry.

This manuscript provides binding epitope information for the 3 mCherry nanobodies and proposed novel tandem multivalent RFP nanobodies, which potentially have significant application prospects. However, before acceptance for publication, there are some issues and problems the authors should address and solve:

1.     Line 17-20, epitope of the binding and kinetic parameters of LaM2, LaM4, and LaM6 are not present in this manuscript. The authors may revise this sentence to describe actually what their analysis on LaM2, LaM4, and LaM6 included. Also, in line 335, the author may add citation after “those developed previously,”.

2.     Line 63 and 65, the authors may not use “traditional antibodies” that is not a professional term;

3.     Line 115, pleases handle “(Error! Reference source not found)” which appears throughout the manuscript;

4.     Line 125-126, the authors should explain how can SDS-PAGE result shows a molar ratio of 1:1 for the 2 proteins complex, any quantification analysis?

5.     Line 145, the authors should explain why does a smaller elution volume of SEC indicates a high affinity between LaM and mCherry?

6.     Figure 2, the authors should modify this figure to better present it, currently it is not optimal to read;

7.     Line 187, Figure 2D should be Figure 2B;

8.     Line 233, chose should be chooses;

9.     Figure 3, A, C, E, G may be deleted;

10.  Line 274 and 287, the authors should add “respectively” when they report multiple values, otherwise reader don’t know which is which;

11.  Line 277, the authors should explain the meaning of “best kdis”;

12.  Line 311-312, the authors should show which data (figure) support the result from this sentence;

13.  Line 358, for the results being reported in this paragraph, the author may make a new table to put together the data for the individual and combined nanobodies, so that readers can easily compare;

14.  Line 361-362, the authors may explain how to tell the maximum thickness from a BLI results; line 377, reported concentrations of mCherry that are assayed in this line are inconsistent with the numbers in the figure A and B;

15.  Fig. S2, it is not easy for readers to locate a specific band, the authors may redesign or simplify this figure.
